

# Interannual Variability in contributions of the Equatorial Undercurrent (EUC) to Peruvian Upwelling

Gandy Maria Rosales Quintana[1], Robert Marsh[2], and Luis Alfredo Icochea Salas[3]

[1]Tokyo University of Marine Science and Technology, Japan
[2]University of Southampton, UK
[3]Universidad Nacional Agraria La Molina, Peru

**Correspondence:** Gandy Maria Rosales Quintana (gandy.rosales@gmail.com)

**Abstract.** Time-varying sources of upwelling waters off the coast of northern Peruvian are analysed in a Lagrangian framework, tracking virtual particles backwards in time for 12 months. Particle trajectories are calculated with temperature, salinity and velocity fields from a hindcast spanning 1988-2007, obtained with an eddy-resolving (1/12º) global configuration of the NEMO ocean model. At 30 and 100 m, where late-December coastal upwelling rates exceed 50 m per month, particles are
seeded in proportion to the upwelling rate. Ensemble maps of particle concentration, age, depth, temperature, salinity and density reveal that a substantial but variable fraction of the particles upwelling off Peru arrive via the Equatorial Undercurrent (EUC). Particles follow the EUC core at around 250 m, characterised by temperatures of around 15-17ºC, salinities in the range 34.9-35.2, and densities of $\sigma$ = 25.5-26.5 . Additional inflows are via two slightly deeper branches further south from the main system, at around 3ºS and 8º. The annual percentage of particles recruited by the EUC (17.5-47% and 16.5-54.6%, from 30 and
100 m respectively) reveal that more of the Peruvian upwelling can be tracked back to the EUC during El Niño and weak La Niña events. In contrast, upwelling waters are of more local origin during a strong La Niña. Annually averaging EUC transport at specific longitudes, a notable negative-to-positive transition is evident during the major El Niño /La Niña events of 1997-99. On short timescales, a degree of longitudinal coherence is evident in EUC transport, with transport anomalies at 160ºW evident at the Galapagos Islands (92ºW) around 30-35 days later. It is concluded that the Peruvian upwelling system is subject to a
variable EUC influence, on a wide range of timescales, most notably the interannual timescale of El Niño Southern Oscillation (ENSO). Identifying this variability as a driver of shifts in population and catch data for several key species, during the study period, these new findings may inform sustainable management of commercially-important fisheries off northern Peru.

## 1   Introduction

A key feature of the tropical Pacific circulation is the Equatorial Undercurrent (EUC) (Cromwell et al., 1954; Knauss, 1959; Lukas, 1986). The EUC originates in the western equatorial Pacific, just north of Papua New Guinea, as an eastward thermocline flow in the depth range 180-280 m at 147ºE. The EUC strengthens eastward, reaching peak transport at around 140ºW



(Knauss, 1959), in a core located within ≈ 3°N and 3°S of the Equator (Blanke and Raynaud, 1997; Johnson et al., 2002; Brown et al., 2007). Near the Galapagos Islands, the EUC core shifts to around 0.5°S (Kessler, 2006; Karnauskas et al., 2010), before
strengthening again towards the eastern boundary (Johnson et al., 2002). This sub-surface flow plays a crucial role in regional climate and biogeochemistry, through substantial transport of nutrient- and carbon-rich cold water to the surface, feeding the so-called "cold tongue" upwelling region (Chavez et al., 1998; Pennington et al., 2006; Chavez and Messié, 2009; Qin et al., 2015; Wang et al., 2019).

Around the global coastline, wind-driven upwelling results in high nutrient supply to the surface layer, enhancing primary
production where light and nutrient levels are optimal. At the eastern boundary of the Atlantic and Pacific basins in each hemisphere, the "Big Four" upwelling systems - Benguela, California, Iberia/Canary and Chile/Peru - are the most active in the world, accounting for approximately 12 out of 17 million metric tons of marine fish catch-year between 2000 to 2007 (representing 20% of the global taken over an area of less than 1% of the global ocean) according to FAO (in Chavez and Messié (2009)). Of the Big Four, the Peruvian part of the Peru/Chile system presents the highest average volume of upwelled
waters (1.6 Sv), even though upwelling-favorable winds are the weakest in average (5.7 m s$^{-1}$) (Chavez and Messié, 2009; Kämpf and Chapman, 2016).

Peruvian upwelling is dynamically linked to the EUC, itself a sub-surface consequence of equatorial surface flows that are driven by easterly trade winds, pushing surface water to the west along the Equator in the South Equatorial Current and creating a pressure head in the western Pacific. Beneath this wind-dominated surface layer, where the Coriolis effect disappears at the
Equator, an eastward pressure gradient drives water back to the east within the EUC, which shoals across the basin from around 250 m in the west to reach the surface in the east Knauss (1959). This equilibrium is interrupted as the trade winds weaken or reverse under during El Niño, with which are associated changes in the quantity and properties of upwelled waters off Peru.

El Niño affects not only the upwelling system, but also brings socioeconomic and cultural impacts to Peru. For instance, in 1982-83 and 1997-98 El Niño events, principal fisheries in Peru such the Peruvian Anchovy, Peruvian hake, giant squid and
others collapsed completely, due to migration and dispersion of the biomass in the region (Ñiquen and Bouchon, 2004; Tam et al., 2006; Wolff et al., 2007). Total pelagic fish landings decreased from 3.3 million tonnes in 1982 to 1.4 millions in 1983 and to almost zero in 1984, as catch species transitioned to Sardine (Arntz and Tarazona, 1989) and others such Jack Mackerel and Pacific Mackerel (Ñiquen and Bouchon, 2004). During the 1997-98 event, the contribution of fisheries to Gross Domestic product (GDP) fell to less than 1%, impacting the employment of approximately 80,000 people (Kämpf and Chapman, 2016).
Moreover, under these anomalously warm conditions, Peru experiences severe flooding and extensive climate-related damage to infrastructure, transport, health, employment, throughout the country, making Peru and other countries in the region highly vulnerable to El Niño.

In the present study, we address the origin and nature of the upwelled waters of one of the most productive upwelling systems in the world. We specifically address the role of the EUC in Peruvian coastal upwelling. We investigate how El Niño and La
Niña conditions thus modulate the Peruvian Coast upwelling system. To proceed, we apply a Lagrangian method for tracking upwelling particles in backwards through time, using an ocean model hindcast spanning 1988-2007. In the following, Section 2 describes the methodology used along with brief details of the numerical model and hindcast. In Section 3, we present a range





of results, to illustrate variability in the source of upwelling waters, and in the EUC itself. In Section 4, we reach conclusions regarding the consequences of El Niño and La Niña for the EUC and Peruvian upwelling, and discuss our findings in relation

to previous studies. Finally, Section 5 presents the Code and data availability.

## 2    METHODOLOGY

We first describe the model that provides the hindcast data needed for the Lagrangian analysis and EUC transport calculations, which are subsequently outlined.

### 2.1    Model Description

We sample 5-day averages of temperature, salinity and velocity data from a hindcast spanning 1988-2007, previously obtained with the Nucleus for European Modelling of the Ocean (NEMO) ocean model (Madec, 2008) in eddy-resolving global configuration (ORCA12), henceforth NEMO-ORCA12. For details of parameterization, initialisation and forcing of this hindcast, see Blaker et al. (2015). The model has a horizontal resolution at the equator of 1/12º (9.277 km), with 75 vertical levels from the surface up to 5902 meters depth, alongside finer grid spacing near the surface (38 levels from 0 to 411 meters). The region

of focus in this study is the South Eastern Pacific, extracted as the region from 170ºW to 75ºW and from 5ºN to 15ºS, for the purposes of Ariane calculations. An advantage of using fields from a fully global model is that remote influences on the region of interest are fully represented, rather than prescribed at the boundaries in a regional model, which can be problematic.

### 2.2    Lagrangian Analysis

To efficiently analyze the provenance of water upwelling off the Peruvian Coast of NEMO-ORCA12, we use the ARIANE

Lagrangian code, based on the original method of Blanke and Raynaud (1997). This mass-preserving numerical Lagrangian approach has proved to be an appropriate method for studying the origin and fate of water masses in a wide range of studies (Doos, 1995; Blanke et al., 2002). Ensembles of particles "seeded" in the coastal upwelling zone off Peru were tracked in "backward" mode, reversing in time the analytical calculation of particle progress through grid cells, to reveal exact pathways.

Allocating particles in proportion to the upwelling rate at three "release depths" (model levels closest to 30 m, 50 m and 100

80    m), we release a variable number of particles per annum, from 669 (e.g., 1991 for 100 m, see Table 2, Figure 15 in annexes) up to 10829 (e.g., 1997 for 30 m, see Table 2, Figure 15 in annexes). Initial locations were evenly located within grid cells on the ORCA12 mesh, where the 5-day averaged upwelling rate exceeded 50 m per month. Calculations based on initial upwelling from the 50-m release depth were relatively similar to those from the 30-m release depth, so we will show results for 30-m and 100-m release depths only.

Particles were released on 31 December of each year and followed back in time for 12 months, to sample inter-annual changes in 3D pathways and water properties (temperature, salinity) of consequence for Peruvian upwelling. Particle data were statistically analyzed on a grid of resolution 0.5º x 0.5º to quantify particle "concentration", dividing the number of particle occurrences passing through each grid square by the total number of particles occurrences during the 1-year tracking period.



For an average representation of main pathways in the study region, we computed a "grand ensemble" for 1988-2007 (for the 30-m and 100-m release depth). Alongside particle concentration, we also average particle age, depth, salinity, temperature and potential density on the 0.5º x 0.5º grid, providing further context for interannual variability of inflow to the Peruvian upwelling system.

## 2.3 Pathways and Transport

Informed by the Lagrangian analysis, we also compute EUC volume transport by integrating eastward flow between 3ºN to 3ºS, encompassing most particle trajectories, and in agreement with previous studies (Blanke and Raynaud, 1997; Johnson et al., 2002; Brown et al., 2007), at selected longitudes from 160ºW to 85ºW, every 10º from 160ºW-100ºW, and at 95ºW, 92ºW and 85ºW respectively. Naturally accommodating EUC flow along shoaling isopycnals, we specifically compute transports binned in potential density, $\sigma$, in the range $\sigma$ = 23.0 - 27.0, at intervals of $\sigma$ = 0.1, using the CDFTOOLS diagnostic package (https://github.com/meom-group/CDFTOOLS). We thus assemble time series of 5-day averaged EUC transport in density space at the selected longitudes.

## 3 RESULT AND DISCUSSION

We begin with a brief evaluation of the NEMO-ORCA12 hindcast in our study region. We then provide an overview of the Lagrangian calculations, before we focus on the pathways followed by water upwelling off Peru, and variation in the source of these waters.

### 3.1 Evaluation of NEMO-ORCA12 hindcast in the equatorial Pacific

To establish whether the NEMO-ORCA12 hindcast realistically simulates the equatorial Pacific circulation, we extract vertical profiles of the monthly mean velocities from the model at longitudes along the Equator (170ºW, 140ºW and 110ºW) corresponding with observational data from NOAA-ADCPs, as shown in Fig. 1. The principal eastward subsurface and westward surface currents, as described in previous studies (Cromwell et al., 1954; Knauss, 1959; Lukas, 1986; Johnson et al., 2002), are clearly reproduced by NEMO/ORCA12. Shoaling of eastward-flowing core of the EUC towards the Galapagos Islands, can be well identified between 50 and 200 m depth with velocities higher than 0.5 m s$^{-1}$ in most of the sections shown in Figure 2. At 140ºW and 110ºW, the EUC core is seen at around 50 to 150 m, with values notably higher than 1 m s$^{-1}$. The highest core velocity along the equator, is found at 140ºW longitude.

### 3.2 Upwelling rates and backtracking on monthly timescales

We define the upwelling off the northern Peruvian coast where upward advection exceeds 1.9x10$^{-5}$ m s$^{-1}$ (50 m per month), for each release depth in the water column, chosen to sample upwelling over a depth range typical of the Peruvian upwelling system (Kämpf and Chapman, 2016). To illustrate vertical velocities at two of our chosen depths, monthly NEMO-ORCA12 climatologies (1988-2007) at 33 m and 100 m are presented in Figures 3 and 4 respectively. Monthly upward velocities are





elevated along both the equatorial upwelling region and off the Peruvian coast, with clear seasonal variability. During austral
summer (December-January-February), upwelling is stronger and more widespread north and south of the Equator, and slightly
weaker off Peru. During austral autumn (March-April-May), equatorial upwelling is characterised as a narrower zone (restricted
equatorward of 3º), with higher values towards the east; off Peru, stronger upwelling is evident, compared to summer. During
austral winter (June-July-August), strong upwelling extends across the easternmost equatorial region (140ºW to Galapagos)
and off Peru, when south-east trade winds are seasonally stronger. A notable difference is evident in the 100-m analysis, with
downwelling stronger around the Galapagos Islands and near the Ecuadorian coast.

Returning to particle release off Peru, and to schematically visualize how these particles interact when released at 30 m, we
illustrate in Fig. 5 the first 30 days of back-tracking for each year of the hindcast. For most of this December-only ensemble,
we observe an eastward origin for some particles, from the Galapagos Islands via the Equator (most clearly represented in
1991,1997, 2004). This is consistent with eastward shoaling of the EUC (to depths below 40 m). However, in other years,
particles originate from a northern shallower region, in the vicinity of the Ecuadorian coast (e.g. 1988, 1989, 1990). For
particles initialised at 100 m, Fig. 6, we observe a similar pattern of inflow at greater depth.

### 3.3   Mean pathways at annual timescale

As outlined in Section 2.2, particle concentration and mean age maps are obtained on a 0.5º x 0.5º mesh for each year from
1988 to 2007. The 20 ensembles are then further averaged to obtain the "grand ensemble" result for 30-m and 100-m release
depths as shown in Fig. 7 right and left panels respectively. Tracking backwards, Fig. 7 show the "grand mean ensemble" for the
corresponding particle concentration, depth, temperature, salinity and potential density ($\sigma$) distributions for flows feeding the
upwelling off Peru, including particle age as contours, every 60 days (see contours in Fig. 7b, left and right panels), for particles
released on 31 December. Ages are expressed as days since 31 December, so day 0 to day 365 runs from 31 December back to 1
January. We use a logarithmic scale for particle concentration, to emphasize a particularly wide range of this diagnostic. In Fig.
7a (left and right), highest particle concentration and youngest age (0-120 days) are naturally located near the release section
(contour lines) in both experiments. As particles spread westward, high concentrations along $\approx$ 3ºN to 3ºS are associated with
sub-surface eastward flow of the EUC, widening slightly west of the Galapagos Islands. For the 30-m release depth, particle
concentration rapidly declines further south and north of 3º to the west of 110ºW, while for 100-m release depth we note
patches of relatively higher particle concentration (compared to Fig. 7a) in the southeast of the region.

Inflows remain at relatively shallow depth (above $\approx$ 120 m) during the first 120 days (spanning December to September
in Fig. 7b, left and right panels) with temperature around 17ºC and salinities between 34.9 and 35.1 along $\sigma = 25.5 - 26.0$
isopycnals (for particles released at 30 m, Fig. 7d left panel). For the same period of time, particles released at 100 m are
characterised with temperatures in the range 15-17ºC and salinities around 34.9 along isopyncnals at $\sigma < 26.5$ (Fig. 7f right
panel). After 240-300 days back-tracking (spanning May to March), particles lie at greater depths (below 240 m) for both
release depths. Along the EUC core, between $\approx$ 1ºN to $\approx$ 1ºS, temperatures fall below 17ºC (Fig. 7d, left and right panels)
and salinities lie between 34.9 – 35.2 (Fig. 7e, left and right panels), combining for $\sigma > 26.5$ isopycnal (Fig. 7f, left and right
panels).





Two relatively deep branches (Fig. 7a, left and right panels) are identified further south of the main equatorial system, along ≈3ºS and ≈8ºS (Fig. 7a, left and right panels) in agreement with previous studies (Lukas, 1986; Johnson and Moore, 1997;

Donohue et al., 2002; Montes et al., 2010), more clearly identified with potential density (i.e. Figure 7f left panel). From both release depths, these southern branches are associated with higher density ($> \sigma \approx 26.5$), with the southern branch (≈8ºS) carrying the densest inflow and the northern branch at somewhat lower density (i.e. Figure 7f left panel), in agreement with previous studies (Montes et al., 2010; Kuntz and Schrag, 2018).

For those particles released at 30 m, the denser southern branch (left panels for Fig. 7d, Fig. 7e and Fig. 7f), is related to high

salinity rather than low temperature ( >35.1, 17ºC or less, and $\sigma$ = 25.5-26.5 ). Somewhat different characteristics were found for particles released at 100-m (right panels for Fig. 7c, Fig. 7d, Fig. 7e and Fig. 7f), for which temperature is a more dominant influence on density. The branch at ≈3ºS is likely associated with a bifurcation of the EUC, to the west of the Galapagos Islands (Johnson et al., 2002; Karnauskas et al., 2010; Jakoboski et al., 2020). This source of Peruvian upwelling is thus indirectly fed by the EUC; this may be particularly the case from April to June, when EUC transport is at its strongest.

## 3.4 EUC transport across the eastern Pacific

Particles released off Peru in late December originate from depths in the range 240-260 m (Fig. 7c, left and right panels), mostly within 3º of the Equator (Fig. 7a, left and right panels), indicating that a substantial proportion of the Peruvian upwelling is recruited from the main EUC system. With this perspective, we compute EUC transport as a function of density in the range $23.0 \leq \sigma \leq 27.0$, at 160ºW, 150ºW, 140ºW, 130ºW, 120ºW, 110ºW, 95ºW, 92ºW and 85ºW - see Fig. 8. In these monthly

transport climatologies (averaged over 1988-2007), EUC flow is associated with highest eastward transport at thermocline densities, in the $\sigma$-range 25.8-26.7. The overall structure of transport in longitude-density space is consistent with observations and other models (Johnson et al., 2002; Qin et al., 2015). From September to December, eastward transport of warm waters by the intensified EUC leads to a flattening of the thermocline. Focussing here on the thermocline density range, westward flows are only evident, for part of the year, at high and low density in Fig. 8.

Integrating positive transports across density classes, for total EUC transport, the monthly climatology in Table 1 shows a degree of seasonality, with peak transport in April from the mid-Pacific (57.1 Sv at 160ºW) to Galapagos Island (23.7 Sv at 92ºW). This seasonality is evident for most years of the hindcast, shown in Fig.9a. The monthly transport anomalies shown in Fig.9b) indicate transport anomalies that can exceed ± 20 Sv during the April peak, notably during the first half of 1990, 1992 and 1993, when negative EUC transport anomalies exceed -20 Sv at 160ºW. Of particular interest here is the development of

negative anomalies during El Niño, followed by positive anomalies during La Niña.

During the historic El Niño of 1997/98, the EUC disappeared from the central Pacific from December 1997 to January 1998, when the West Pacific Warm Pool migrated eastward with the collapse of the trade winds and the equatorial cold tongue (McPhaden, 1999). Transport anomalies remarkably exceeded -20 Sv for much of 1997 (Fig.9b). Other notable periods of sustained negative EUC transport anomalies span the end of 2002 and beginning of 2003, and late 2006. All of these events

were succeeded by strongly positive anomalies, from the most western region (160ºW) to the east, most strikingly over 1997-99 (Fig. 9b), when the El Niño transitioned to a sustained La Niña.





According to previous classifications, major El Niño/La Niña events are defined as an intense warming (cooling) over most of the equatorial Pacific, with the strongest oceanic signature located in the eastern/central region (McPhaden, 1999; Takahashi and Dewitte, 2015; Bertrand et al., 2020). Notable recent events are the 1982/83 and 1997/98 El Niños, the 1998/99 La Niña, and the 2015/16 El Niño. Moderate Eastern (80-90ºW, 5ºN-5ºS) or Central (160ºE-150ºW, 5ºN-5ºS) El Niños are defined as modest equatorial warming with strongest/weak oceanic signature off the South American coast (Yu et al., 2011). Moderate Eastern El Niños are apparent in 1991/92 and 2006/07, whilst the 2000/01 La Niña may be categorised as a weak Eastern event (Yu et al., 2011).

Given the initial (total) number of upwelled particles off the north Peruvian coast (where upwelling exceeds 50 m per month) and the total number of particles crossing specific locations along the equatorial Pacific, we calculate percentage recruitment from the EUC (3ºN -3ºS), shown in Fig. 10. During the major El Niño of late 1997, a substantial fraction of particles tracked from the Peruvian upwelling zone crossed 160ºW in the EUC within the year. Evidently, the EUC contribution to Peruvian upwelling is most dominant during the major El Niño of 1997-98 (Fig. 9, Fig. 10). Full details of the percentages of upwelling particles recorded earlier in the EUC at selected longitudes are provided in Tables A1 and A2.

For the 30-m (100-m) release depth, the highest percentage of particles found in 1997 varied from 39.5% (34.4%) at 160ºW to 86.6% (93.2%) at 92ºW. This is consistent with flattening of the thermocline with the onset of El Niño in 1997 (McPhaden, 1999), which limited the upwelling of EUC waters in the central Pacific and allowed more of these waters to progress all the way to the eastern boundary, where upwelling continued along the Peruvian coast. Conversely, during La Niña, smaller percentages of upwelling particles arrived via the EUC. Lowest values in 1998 varied from 0.0% (2.3%) at 160ºW to 27.7% (46.4%) at 92ºW, for 30-m (100-m) release depth (see Fig. 10, Tables A1 and A2). This reduced contribution of the EUC to Peruvian upwelling paradoxically coincides with anomalously strong transports at 160ºW during late 1998, which persist into 1999 (transport and anomalies >50 Sv and >20 Sv respectively, see Fig. 9). However, the lowest percentages of upwelling particles in the longitude range 120-160ºW during La Niña supports a hypothesis that only particles reaching an easternmost extension of the EUC are likely to influence Peruvian upwelling, consistent with evidence from 1997 and other El Niño events (Firing et al., 1983; Halpern, 1987; Mcphaden et al., 1990; Johnson et al., 2000).

A second highest peak, for the number of upwelling particles and percentage associated with the EUC, happened during the Eastern Weak La Niña of 2000/01 (Fig. 9). In 2000, the percentage of particles arriving via the EUC varied between 35.2% (26.5%) and 76.6% (73.5%) at 160ºW and 92ºW for 30-m (100-m) released depth. By 2001, recruited from the west was substantially reduced, with only 5.9% (12.8%) present at 160ºW. Enhanced EUC influences on Peruvian upwelling late in 1997 and 2000 are a consequence of very different variability. During the strong El Niño of 1997, the EUC was severely weakened but conveyed particles all the way to the eastern boundary. In contrast, during the Eastern Weak La Niña of 2000, EUC strengthening extended at least to 92ºW (see Fig. 9), conveying more particles eastward. Rather than upwelling along the Equator, as during a strong La Niña, more EUC water reached the eastern boundary during this weak La Niña.



### 3.5 Impacts on Peruvian upwelling linked to variable EUC contributions

These episodes of enhanced recruitment from the western basin likely impact Peruvian upwelling. Both the 1997/98 and 2000/01 events had important effects in fisheries. May-June 1997 fieldwork conducted by the Peruvian Sea Institute (IMARPE), revealed an unusual southern and deeper (> 150-200 m) migration of Peruvian Hake (Merluccius gayi peruanus; generally distributed between 0ºN to 8ºS), easily reaching 12ºS and even more southern areas along the Peruvian coast (Castillo et al., 1997). Conversely, Hake tended to migrate northward and sometimes closer to the surface, due bottom oxygen deficiency,

during the La Niña of 1998/99. This is consistent with the biogeochemical character of the EUC. Where a southward branch of the EUC reaches the northern Peruvian coast (Fig. 7), it refreshes the sub-surface layer with high concentrations of dissolved oxygen (Echevin et al., 2020) and other vital nutrients. A similar impact afflicts the second most abundant demersal species in Peruvian coastal waters, the Conger eel (Ophichthus remiger). During recent El Niño events (December 1997 to May 1998; March-July 1992) and subsequent La Niña events (November 1993 to March 1994; April-August 1998), significant enhanced

and reduced Catch per Unit Effort (CPUE) was reported, respectively (Castillo et al., 2000; Martina, 2004). The oceanographic and fisheries evidence together suggest crucial links between demersal species and the EUC. More recently, Martina (2018) also highlighted dramatic reduction in biomass during 2004/05, followed by significant increase in 2007. The latter increase coincides with positive EUC transport anomalies taking place during early 2007 (see Fig. 9b).

During the moderate Eastern Pacific El Niño of 1991/92, an alternative transition is observed. A relatively high percentage of

235 upwelling particles is traced back to the EUC, with 31.4% (9.4%) and 53.2% (57.4%) at 160ºW and 92ºW respectively during 1991, for 30-m (100-m) release depths (Fig. 10b and Fig. 10d). These enhanced percentages of EUC origin coincide with relatively small anomalies in eastward transport (Fig. 9b). Through 1992, particles from 30-m release depth were present in much lower percentages along the EUC, while particles back-tracked from 100-m appeared in higher percentages (see Tables A1 and A2), consistent with a subsequently deeper thermocline and EUC during 1992. Related to the 1991/92 El Niño was a

240 dramatic size-age and biomass reduction in 1992, along with spatial migration (Castillo, 1996), as during the major El Niño of 1997. These changes were previously attributed to a combination of overfishing by foreign fishing vessels during 1989-90 and sub-optimal oceanographic conditions during the 1991/92 El Niño, when the EUC disappeared from most of the eastern Pacific. While species are often slow to recover in size-age-abundance indicators, even after the major El Niño of 1997/98, recovery is observed in CPUE data as wider size structure and higher catch numbers.

In summary, the EUC contribution to upwelling along the northern Peruvian coast is sometimes substantial, altering environmental conditions of consequence for many marine species. It has further been suggested that some species also use the EUC as a migration conveyor, such as the Pacific Giant squid (Dosidicus gigas). Largely absent from east Pacific coastal waters over 1950-90, catches of this species increased from the beginning of 1991 through 1992. During 1995-96, overfishing impacted this population, primary due to the high catch effort of foreign fishing vessels (Contreras Paya, 2017). Subsequent to the El Niño,

of 1997-98, a denser nearshore migration of the squid, more accessible for fishing, was observed off the North American coast (CALCOFI, 2000).Changes in coastal squid populations may be linked to the equatorially and then coastally trapped Kelvin



waves (e.g., Shaffer et al. (1997)), which may favour southward (northward) migration when the EUC is stronger (weaker) around the Galapagos Island, as the squid seek out optimal conditions (Cheung W. L. et al., 2018).

We identify less intense events at other times, such late 2002 to early 2003 (Fig. 10b and Fig. 10d), when Peruvian Hake

almost disappeared from a fishery off northern Peruvian, leading to a moratorium on fishing this species until March 2004 (Lassen et al., 2009; Benites and Barriga, 2011). Following the moderate Eastern El Niño of 2006-07, a substantial percentage of particles upwelling at 100-m in late 2007 are traced back to the EUC, in contrast to very few particle back-traced from 30-m, suggesting that the EUC remained anomalously deep following the El Niño.

### 3.6 Longitudinal coherence of EUC transport anomalies

To examine the longitudinal coherence of EUC transport anomalies, we compute correlation between EUC transport at each selected longitude (Table 2). Despite generally low values, we find significant (p-value< 0.01, 99% confidence interval) positive correlations at zero-lag. This is consistent with rapid and near-simultaneous variations in EUC transport across the eastern Pacific, although weakening of correlation with separation (in longitude) indicates that transport anomalies may advect eastward, associated with inertial terms in the momentum balance and eastward propagation of Kelvin waves. The development of

anomalous flow to the east may thus be predictable for an appropriate time lag. Experimenting with lagged cross correlations of transport at 160ºW and 92ºW, we obtain highly significant R = +0.31 (p-value < 0.01) between transport at 92ºW and transport 30-35 days earlier at 160ºW. With further investigation, beyond scope here, more skilful EUC predictability may provide useful advance warning of substantial changes in the Peruvian upwelling system, and subsequent impacts on important marine resources. A further step would be assess the variability and predictability of year-round inflows to the Peruvian upwelling

system, also beyond the scope of this preliminary investigation.

### 4 Conclusions

We have systematically quantified the origin of waters upwelling in one of the most productive regions in the global ocean, the Peruvian Upwelling System. Analyses are based on a model hindcast spanning 1988-2007, which includes major warm and cold events associated with El Niño and La Niña respectively. Through Lagrangian analysis of virtual particle ensembles, we

established that an interannually variable fraction of the particles is recruited via the Equatorial Undercurrent (EUC), flowing between ≈ 3ºN to 3ºS at around 250 m. A key inference is that the northern Peruvian upwelling system is sensitive to highly variable EUC inflow.

Particle back-trajectories - sampling the upwelling at depths of 30-m and 100-m off Peru - trace the EUC as far west as 170ºW, moving at depths, temperatures, salinities and densities that are consistent with observations (Johnson et al., 2002).

Back-trajectories further identified two relatively deep branches south of the main equatorial system, along ≈3ºS and ≈8ºS, in agreement with previous studies (Lukas, 1986; Johnson and Moore, 1997; Donohue et al., 2002; Montes et al., 2010). Only a small percentage of particles otherwise originate from latitudes poleward of 3º, as far west as 170ºW.

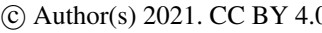

Over the 1988-2007 hindcast period, we quantified the variable contribution of EUC waters to Peruvian upwelling in late December. At 92ºW, we identify highest and lowest percentages of EUC-sourced particles during the strong El Niño of 1997 and the subsequent strong La Niña of 1998, respectively. The EUC is thus most influential when the equatorial thermocline is flattened during El Niño, allowing flow all the way to the eastern boundary. During a strong La Niña, EUC waters conversely upwell with a shoaling thermocline to reach the surface layer in the central Pacific, far to the west of the eastern boundary. In this scenario, the waters upwelling off Peru are of local provenance.

Variable provenance of oxygenated, nutrient-rich EUC waters in the 1990s can be linked to substantial changes in Peruvian coastal fisheries. Further variability in the 2000s is associated with a range of El Niño and La Niña events. On sub-annual timescales, particles most rapidly cross the eastern Pacific with the EUC during peak transport around March and April, consistent with previous studies (Flores et al., 2009). Correlations between 5-day averaged EUC transport at selected longitudes in the range 92-160ºW indicate a high degree of longitudinal coherence, with evidence of a time lag of 30-35 days for transport anomalies at 160ºW to reach the Galapagos Islands at 92ºW.

In highlighting the impact of variable EUC influences on regional biogeochemistry, ecosystems and fisheries, this study provides the basis for informed analysis and prediction of an unfolding El Niño or La Niña, for a sustainable approach to management of marine resources in the Peruvian upwelling system. A next step would be to include biogeochemical analyses, using models and observations, to better understand the consequences of variable nutrient supply for primary productivity at the base of the food chain.

## 5 Code and data availability

The NEMO-ORCA12 data analysed here is archived at the National Oceanography Centre, Southampton. The original version of ARIANE software used here, available from http://stockage.univ-brest.fr/ grima/Ariane/, was adapted at the National Oceanography Centre and the University of Southampton. Specific trajectory data and NEMO-ORCA12 diagnostics presented here are available from the authors, on request. Equatorial current data from the TAO/TRITON Array are available from NOAA via https://www.pmel.noaa.gov/gtmba/pmel-theme/pacific-ocean-tao.

*Author contributions.* .

All authors have contributed equally to this paper.

*Competing interests.* .

The authors declare that they have no conflict of interest.



*Acknowledgements.* We are grateful to the European Mundus Joint Master Degree (ERASMUS MUNDUS Scholarship) and the Marine
Environment and Resources program (MER+) for the financial support provided during this 2-years project. We acknowledge the National
Oceanographic Centre Southampton (NOCS) for undertaking the NEMO-ORCA12 simulation (using the NEMO framework developed by a
consortium of European institutions), and Jeff Blundell at the University of Southampton, for development and installation of a local version
of the ARIANE software that is central to this study.





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





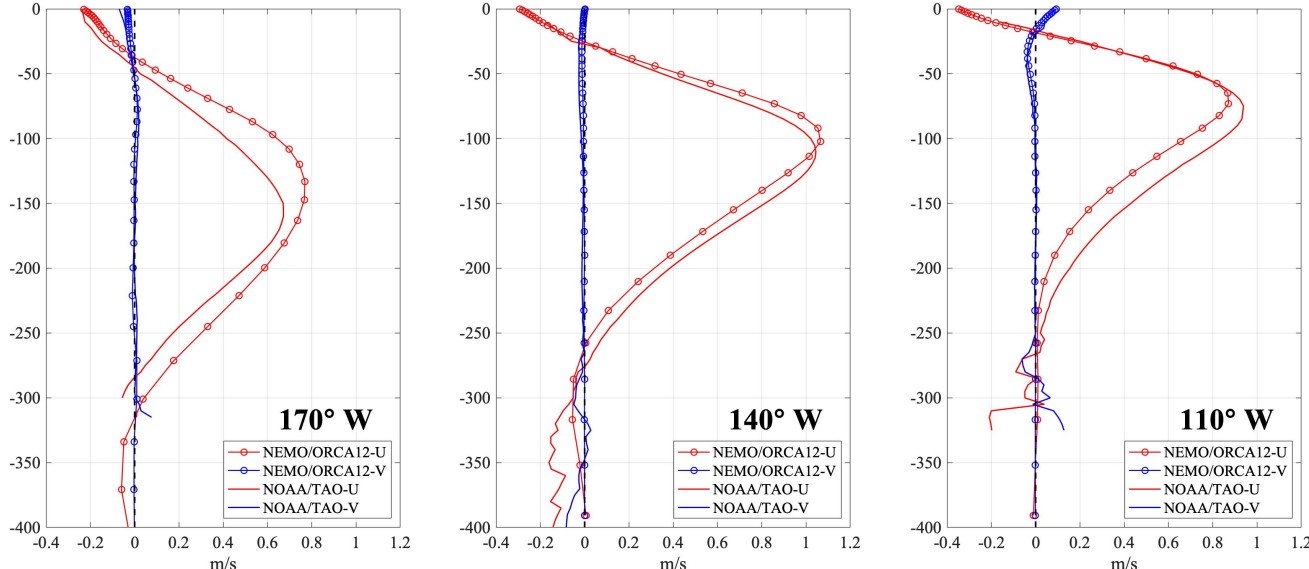

**Figure 1.** Vertical profiles of zonal and meridional components of current (red and blue respectively) averaged over 1988-2007, in the NEMO-ORCA12 hindcast and from NOAA mooorings, on the Equator at 170ºW, 140ºW and 110ºW.





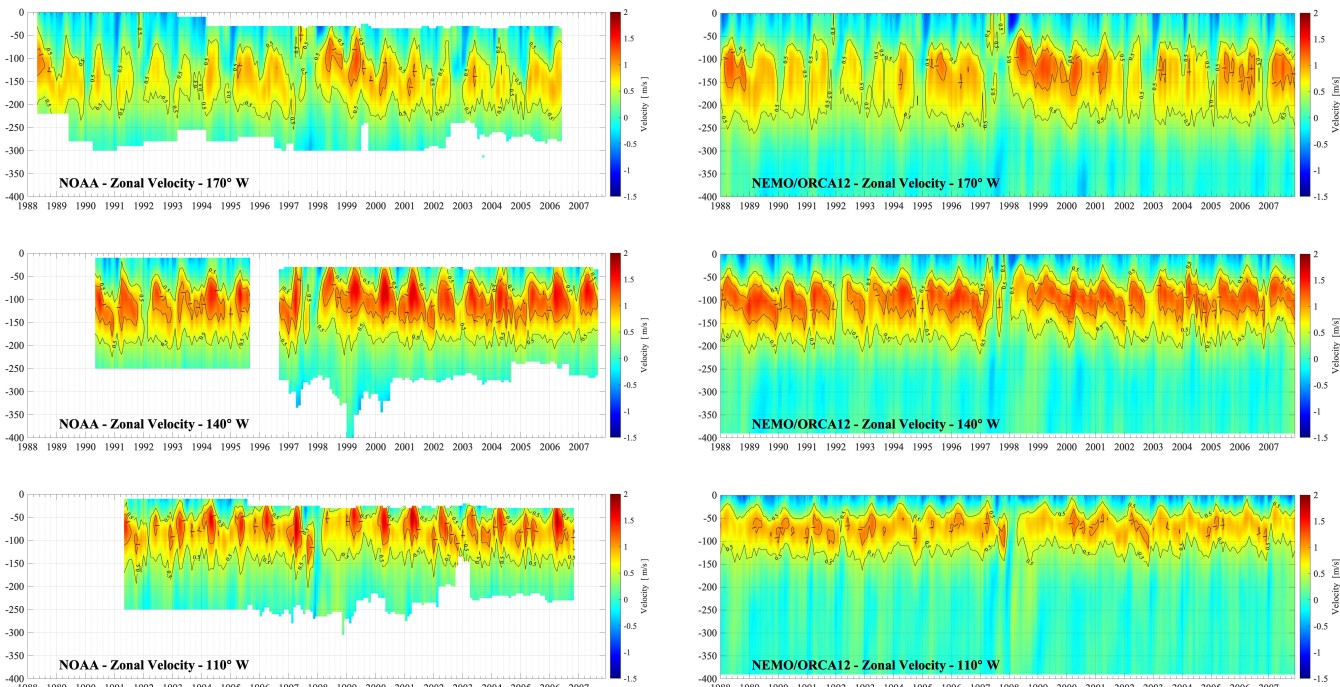

**Figure 2.** Monthly zonal component of current along the equatorial Pacific at 170ºW, 140ºW and 110ºW, from the NEMO-ORCA12 hindcast and NOAA/TAO in-situ data. Contour lines highlight 0.5 m s$^{-1}$ and 1 m s$^{-1}$

.





**Figure 3.** Monthly climatology (averaged over 1988-2007) of vertical velocity at a depth of 33 m in the eastern Pacific, in the NEMO-ORCA12 hindcast.





**Figure 4.** Monthly climatologies (averaged over 1988-2007) of vertical velocity at a depth of 100 m in the eastern Pacific.





**Figure 5.** Ensembles of particles "seeded" off Peru and Ecuador at 30-m depth (initial position in black, where upwelling rates exceed $1.9\times$ $10^{-5}$ m s$^{-1}$ or 50 m per month), tracked backward over 31 days from initial positions (black dots), in each year of 1988 to 2007. Particles are color-coded for depth in the range 0-150 m.





**Figure 6.** As Fig. 5, but for particles seeded at 100-m depth, now specifying particle depth in the range 0-240 m (blue).





**Figure 7.** Grand ensemble of back-trajectory data (averaging the 1988-2007 ensembles), for particles back-tracked from 31 December at 30-m depth (left panel) and 100-m depth (right panel) respectively, binning at 0.5º x 0.5º resolution: (a) particle concentration (calculated as a fraction, by dividing the number of particle occurrence per 0.5º x 0.5º grid cell by the total number of particle occurrences); (b) particle age (days, backward in time); (c) particle depth (m); (d) particle temperature (ºC), (e) particle salinity (psu); (f) particle potential density ($\sigma$, kg m$^{-3}$).





**Figure 8.** Climatological monthly (1988-2007 mean) transport (Sv) between 3ºN-3ºS, in the density range $\sigma = 23.0 - 27.0$ (at 0.1 resolution) at 160ºW, 150ºW, 140ºW, 130ºW, 120ºW, 110ºW, 95ºW, 92ºW and 85ºW along the equatorial Pacific (see Sect. 2.3 for details of these transport calculations.



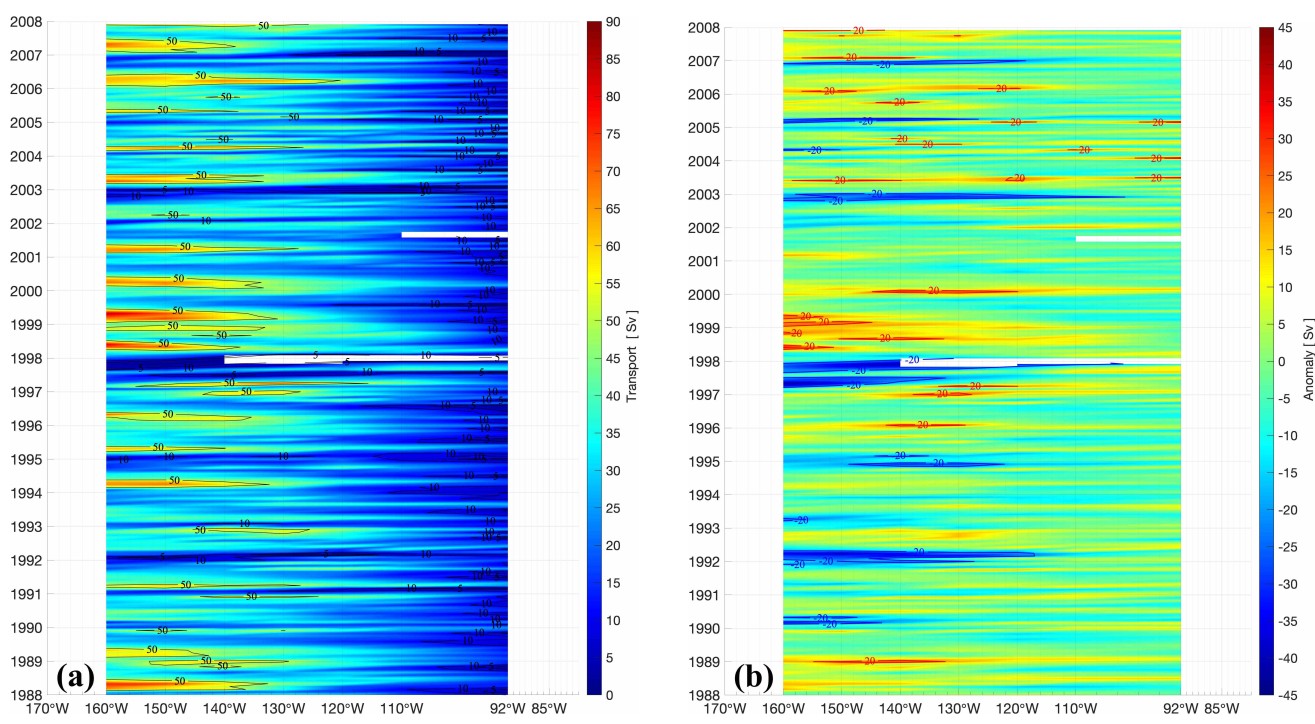

**Figure 9.** Integrated EUC transport based on calculations at 160ºW, 150ºW, 140ºW, 130ºW, 120ºW, 110ºW and 92ºW (Galapagos Islands), in the latitude range 3ºN-3ºS: (a) monthly averages; (b) monthly anomalies.

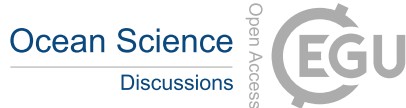



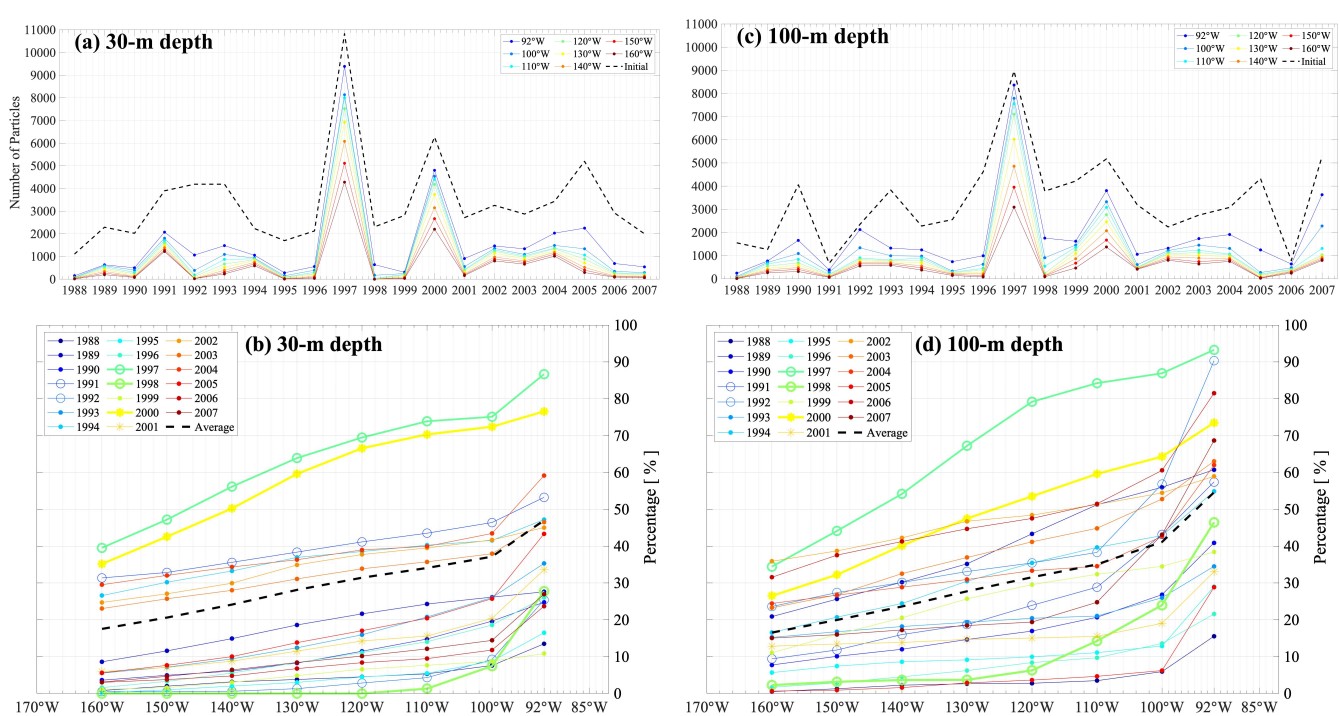

**Figure 10.** Annual number of particles and percentage associated to the EUC (3ºN-3ºS) at 160ºW, 150ºW, 140ºW, 130ºW, 120ºW, 110ºW, 100ºW and 92ºW, for 30-m releases in (a) and (b), and for 100-m releases in (c) and (d).





| Monthly Climatology | 160ºW | 150ºW | 140ºW | 130ºW | 120ºW | 110ºW | 92ºW |
|---|---|---|---|---|---|---|---|
| January | 26.0 | 30.9 | 33.6 | 33.8 | 25.6 | 15.9 | 12.4 |
| February | 30.3 | 27.2 | 21.3 | 20.2 | 19.4 | 15.0 | 6.8 |
| March | 47.0 | 47.6 | 43.9 | 36.3 | 24.0 | 19.0 | 10.7 |
| April | 57.1 | 58.6 | 55.2 | 44.1 | 34.4 | 30.1 | 23.7 |
| May | 58.6 | 55.8 | 43.2 | 36.8 | 30.2 | 27.1 | 17.8 |
| June | 45.0 | 42.0 | 36.2 | 31.6 | 27.5 | 23.5 | 16.3 |
| July | 39.8 | 35.8 | 30.2 | 23.9 | 20.7 | 19.1 | 5.9 |
| August | 31.4 | 29.1 | 25.6 | 23.0 | 21.9 | 18.2 | 12.6 |
| September | 26.2 | 25.8 | 27.2 | 27.4 | 31.8 | 27.6 | 17.3 |
| October | 25.5 | 27.6 | 27.8 | 29.3 | 30.4 | 26.9 | 18.6 |
| November | 29.0 | 35.4 | 38.1 | 37.6 | 34.0 | 26.2 | 7.7 |
| December | 32.6 | 38.5 | 43.0 | 40.6 | 36.8 | 30.2 | 18.4 |
| AVERAGE | 36.3 | 36.4 | 34.6 | 31.5 | 26.8 | 21.5 | 12.6 |

**Table 1.** Monthly Climatology (from 5-day averaging between 1988 to 2007) and annual average for the EUC transports (Sv) at 160ºW, 150ºW, 140ºW, 130ºW, 120ºW, 110ºW, and 92ºW.





| R | 160ºW | 150ºW | 140ºW | 130ºW | 120ºW | 110ºW | 92ºW |
|---|---|---|---|---|---|---|---|
| 160ºW | 1.00 | 0.79 | 0.53 | 0.27 | N/S | N/S | -0.17 |
| 150ºW | 0.79 | 1.00 | 0.71 | 0.45 | 0.23 | N/S | -0.14 |
| 140ºW | 0.53 | 0.71 | 1.00 | 0.68 | 0.42 | 0.19 | -0.11 |
| 130ºW | 0.27 | 0.45 | 0.68 | 1.00 | 0.68 | 0.39 | N/S |
| 120ºW | N/S | 0.23 | 0.42 | 0.68 | 1.00 | 0.70 | 0.20 |
| 110ºW | N/S | N/S | 0.19 | 0.39 | 0.70 | 1.00 | 0.43 |
| 92ºW | -0.17 | -0.14 | -0.11 | N/S | 0.20 | 0.43 | 1.00 |

**Table 2.** Correlation coefficient (R) matrix at 0-lag for the EUC transport at the selected longitudes, for R values with p-values <0.01 (99 % confidence interval). Not significant R values are indicated as N/S.





| Year | Initial | 160ºW | 150ºW | 140ºW | 130ºW | 120ºW | 110ºW | 100ºW | 92ºW |
|------|---------|-------|-------|-------|-------|-------|-------|-------|------|
| 1988 | 1107 | 0.8 (9) | 2.0 (22) | 3.2 (35) | 3.8 (42) | 4.5 (50) | 5.3 (59) | 7.6 (84) | 13.5 (149) |
| 1989 | 2291 | 8.6 (197) | 11.6 (265) | 14.9 (341) | 18.6 (426) | 21.6 (495) | 24.3 (556) | 26.1 (599) | 27.6 (632) |
| 1990 | 2023 | 3.6 (73) | 4.9 (100) | 6.1 (123) | 8.3 (167) | 11.5 (232) | 14.8 (299) | 19.6 (397) | 24.8 (501) |
| 1991 | 3897 | 31.4 (1222) | 32.8 (1280) | 35.6 (1387) | 38.4 (1495) | 41.1 (1603) | 43.5 (1695) | 46.4 (1808) | 53.2 (2074) |
| 1992 | 4188 | 0.5 (21) | 0.5 (21) | 0.6 (25) | 1.3 (55) | 2.8 (119) | 4.3 (180) | 9.2 (387) | 25.4 (1063) |
| 1993 | 4188 | 5.5 (232) | 7.1 (299) | 9.4 (393) | 12.4 (519) | 15.9 (667) | 20.7 (869) | 26.0 (1089) | 35.3 (1474) |
| 1994 | 2232 | 26.6 (593) | 30.2 (674) | 33.2 (742) | 36.9 (824) | 38.5 (860) | 40.2 (898) | 41.5 (927) | 47.2 (1053) |
| 1995 | 1699 | 0.2 (3) | 1.0 (17) | 2.0 (34) | 3.0 (51) | 4.5 (76) | 5.5 (93) | 8.8 (149) | 16.5 (280) |
| 1996 | 2127 | 1.6 (34) | 3.3 (70) | 5.5 (118) | 8.4 (178) | 11.1 (236) | 14.1 (299) | 18.6 (395) | 26.0 (554) |
| 1997 | 10829 | 39.5 (4281) | 47.2 (5111) | 56.1 (6079) | 63.9 (6918) | 69.5 (7523) | 73.9 (7999) | 75.1 (8132) | 86.6 (9383) |
| 1998 | 2308 | 0.0 (0) | 0.0 (0) | 0.0 (0) | 0.0 (0) | 0.0 (0) | 1.3 (30) | 7.5 (172) | 27.7 (640) |
| 1999 | 2810 | 1.1 (31) | 1.6 (45) | 3.0 (85) | 4.8 (136) | 6.6 (185) | 7.7 (215) | 8.8 (246) | 10.8 (304) |
| 2000 | 6269 | 35.2 (2204) | 42.6 (2668) | 50.2 (3148) | 59.6 (3736) | 66.5 (4172) | 70.3 (4407) | 72.4 (4538) | 76.6 (4799) |
| 2001 | 2714 | 5.9 (161) | 7.1 (192) | 8.8 (238) | 11.5 (312) | 14.3 (387) | 15.6 (423) | 20.4 (553) | 33.6 (911) |
| 2002 | 3254 | 24.7 (804) | 27.0 (880) | 29.9 (974) | 34.9 (1135) | 37.7 (1227) | 39.5 (1285) | 41.7 (1356) | 45.0 (1465) |
| 2003 | 2872 | 23.1 (662) | 25.7 (738) | 28.0 (804) | 31.1 (893) | 33.8 (972) | 35.7 (1026) | 37.9 (1089) | 46.5 (1336) |
| 2004 | 3431 | 29.5 (1013) | 32.1 (1100) | 34.3 (1178) | 36.3 (1245) | 39.0 (1337) | 39.9 (1370) | 43.4 (1490) | 59.1 (2029) |
| 2005 | 5197 | 5.6 (291) | 7.6 (397) | 10.0 (520) | 13.8 (717) | 17.0 (886) | 20.4 (1061) | 25.7 (1337) | 43.3 (2251) |
| 2006 | 2919 | 3.0 (88) | 3.8 (110) | 4.8 (139) | 6.7 (197) | 8.4 (244) | 9.5 (276) | 11.8 (344) | 23.7 (691) |
| 2007 | 1999 | 3.1 (62) | 4.7 (93) | 6.4 (128) | 8.4 (168) | 10.2 (203) | 12.1 (242) | 14.4 (288) | 27.0 (539) |
| G. E. | | 17.5 | 20.6 | 24.1 | 28.1 | 31.4 | 34.1 | 37.1 | 47.0 |

**Table A1.** Total percentage (%) and annual number of particles (in brackets) associated with the EUC (3ºN-3ºS), for particles released at 30-m depth. The initial number of particles released off the Peruvian coast is also shown (Initial). The percentage for the grand ensemble (G.E.) is computed from the 20-year hindcast experiment.





| Year | Initial | 160ºW | 150ºW | 140ºW | 130ºW | 120ºW | 110ºW | 100ºW | 92ºW |
|------|---------|-------|-------|-------|-------|-------|-------|-------|------|
| 1988 | 1552 | 0.5 (8) | 1.3 (20) | 2.3 (35) | 2.6 (41) | 2.8 (43) | 3.5 (54) | 5.9 (92) | 15.5 (241) |
| 1989 | 1265 | 20.9 (264) | 25.6 (324) | 30.2 (382) | 35.2 (445) | 43.3 (548) | 51.2 (648) | 55.9 (708) | 60.7 (768) |
| 1990 | 4057 | 7.7 (314) | 10.1 (409) | 12.0 (487) | 14.6 (594) | 16.9 (687) | 20.7 (839) | 26.8 (1088) | 40.9 (1659) |
| 1991 | 669 | 9.4 (63) | 11.8 (79) | 15.9 (107) | 18.7 (125) | 23.9 (160) | 28.9 (193) | 43.2 (289) | 57.4 (384) |
| 1992 | 2351 | 23.6 (555) | 27.4 (644) | 30.1 (708) | 33.1 (778) | 35.4 (833) | 38.3 (900) | 56.8 (1336) | 90.3 (2123) |
| 1993 | 3231 | 15.2 (583) | 16.8 (642) | 18.2 (696) | 19.3 (740) | 20.4 (782) | 21.0 (805) | 25.9 (995) | 34.5 (1322) |
| 1994 | 2278 | 16.6 (377) | 20.7 (472) | 24.4 (556) | 30.5 (694) | 35.4 (807) | 39.6 (903) | 42.8 (976) | 54.8 (1249) |
| 1995 | 2555 | 5.7 (145) | 7.4 (190) | 8.6 (220) | 9.1 (233) | 9.9 (254) | 11.1 (283) | 12.9 (329) | 29.0 (741) |
| 1996 | 4618 | 1.7 (78) | 2.9 (133) | 4.5 (209) | 6.2 (287) | 8.4 (387) | 9.7 (447) | 13.6 (627) | 21.6 (995) |
| 1997 | 8966 | 34.4 (3088) | 44.1 (3956) | 54.2 (4857) | 67.2 (6026) | 79.2 (7098) | 84.2 (7551) | 86.9 (7791) | 93.2 (8360) |
| 1998 | 3784 | 2.3 (86) | 3.2 (120) | 3.7 (138) | 3.7 (140) | 6.3 (237) | 14.2 (539) | 23.9 (907) | 46.4 (1757) |
| 1999 | 4217 | 11.1 (466) | 16.0 (675) | 20.5 (866) | 25.7 (1083) | 29.6 (1246) | 32.4 (1364) | 34.5 (1455) | 38.4 (1619) |
| 2000 | 5174 | 26.5 (1371) | 32.2 (1668) | 40.1 (2076) | 47.5 (2456) | 53.5 (2769) | 59.6 (3083) | 64.3 (3326) | 73.5 (3801) |
| 2001 | 3194 | 12.8 (410) | 13.4 (429) | 13.8 (442) | 14.7 (469) | 15.0 (480) | 15.5 (495) | 19.0 (608) | 33.0 (1056) |
| 2002 | 2238 | 35.9 (803) | 38.7 (866) | 42.2 (945) | 46.7 (1046) | 48.4 (1083) | 51.3 (1148) | 54.4 (1218) | 58.9 (1319) |
| 2003 | 2748 | 23.3 (639) | 26.8 (736) | 32.5 (894) | 36.9 (1014) | 41.2 (1131) | 44.8 (1232) | 52.8 (1450) | 63.0 (1732) |
| 2004 | 3081 | 24.4 (753) | 26.7 (822) | 28.9 (889) | 30.9 (953) | 33.3 (1027) | 34.5 (1064) | 42.7 (1314) | 61.9 (1909) |
| 2005 | 4325 | 0.7 (28) | 0.9 (39) | 1.6 (69) | 2.9 (124) | 3.6 (157) | 4.7 (201) | 6.2 (269) | 28.8 (1246) |
| 2006 | 783 | 31.6 (247) | 37.6 (294) | 41.3 (323) | 44.7 (350) | 47.5 (372) | 51.5 (403) | 60.5 (474) | 81.5 (638) |
| 2007 | 5289 | 15.0 (795) | 15.9 (845) | 17.2 (912) | 18.4 (975) | 19.4 (1025) | 24.8 (1309) | 43.1 (2280) | 68.6 (3630) |
| G. E. |  | 16.5 | 19.9 | 23.6 | 27.7 | 31.5 | 35.0 | 41.1 | 54.6 |

**Table A2.** Total percentage (%) and annual number of particles (in brackets) associated with the EUC (3ºN-3ºS), for particles released at 100-m depth. The initial number of particles released off the Peruvian coast is also shown (Initial). The percentage for the grand ensemble (G.E.) is computed from the 20-year hindcast experiment.





| Year | 160ºW | 150ºW | 140ºW | 130ºW | 120ºW | 110ºW | 92ºW |
|---|---|---|---|---|---|---|---|
| 1988 | 45.2 | 44.5 | 40.7 | 33.9 | 27.6 | 20.8 | 10.6 |
| 1989 | 40.3 | 40.3 | 37.1 | 34.1 | 27.6 | 21.5 | 11.9 |
| 1990 | 31.4 | 32.7 | 33.3 | 31.3 | 27.1 | 22.6 | 11.8 |
| 1991 | 30.8 | 32.6 | 31.3 | 30.3 | 26.9 | 22.1 | 14.4 |
| 1992 | 28.9 | 28.9 | 28.7 | 28.8 | 26.3 | 21.8 | 16.2 |
| 1993 | 30.2 | 29.7 | 29.3 | 27.8 | 24.1 | 19.2 | 13.0 |
| 1994 | 33.9 | 36.1 | 34.0 | 28.8 | 23.6 | 18.1 | 10.3 |
| 1995 | 35.6 | 35.1 | 32.7 | 29.4 | 23.7 | 17.5 | 9.7 |
| 1996 | 38.8 | 38.4 | 36.9 | 32.9 | 26.8 | 19.5 | 11.0 |
| 1997 | 19.3 | 21.4 | 24.3 | 24.9 | 24.7 | 25.2 | 18.2 |
| 1998 | 45.2 | 42.3 | 37.7 | 33.1 | 28.9 | 26.1 | 13.2 |
| 1999 | 48.1 | 45.3 | 39.2 | 33.1 | 26.0 | 18.4 | 9.0 |
| 2000 | 40.9 | 40.5 | 37.4 | 34.1 | 29.2 | 23.7 | 12.2 |
| 2001 | 37.8 | 39.0 | 34.7 | 30.0 | 26.2 | 20.1 | 9.8 |
| 2002 | 26.0 | 28.4 | 28.9 | 28.1 | 26.4 | 22.5 | 12.4 |
| 2003 | 38.1 | 37.0 | 36.3 | 31.7 | 27.5 | 21.8 | 14.2 |
| 2004 | 34.7 | 37.0 | 36.3 | 32.2 | 28.6 | 24.4 | 15.4 |
| 2005 | 36.0 | 36.1 | 35.5 | 32.4 | 26.2 | 20.1 | 11.7 |
| 2006 | 37.7 | 39.1 | 36.1 | 33.2 | 28.5 | 22.0 | 12.0 |
| 2007 | 46.7 | 44.4 | 40.2 | 33.6 | 26.8 | 20.5 | 11.9 |
| AVERAGE | 36.3 | 36.4 | 34.5 | 31.2 | 26.7 | 21.4 | 12.5 |

**Table A3.** Annual EUC transport (Sv) along the Equatorial Pacific at 160ºW, 150ºW, 140ºW, 130ºW, 120ºW, 110ºW, and 92ºW for particles released at 30-m depth.