# Peer review of "Interannual Variability in contributions of the Equatorial Undercurrent (EUC) to Peruvian Upwelling source water"

_Ocean Science, 2021_

## Referee Comment (RC3)

**Review OS-2021-13**

**Major points:**

This is an interesting paper addressing the relationship between the equatorial undercurrent to the upwelling off the coast of Peru.

The objectives of the paper are reasonably clear, in particular, the Lagrangian particle tracking as applied to 20 years of regional data from a high resolution global model to investigate the above relationship.

The focus of the paper is on interannual variability. Though I do realise the interannual variability is large in this region, I did think a more logical approach would be to discuss the mean annual cycle ( 20 year average of each month) first to establish particle tracking and its interpretation. Though this may have been covered in previous published studies your results are based on a very high resolution model and this is new. Therefore it would be better to introduce these results first before the interannual results. The proposed section would lead the reader to an understanding of the particle tracking method and its interpretation. The results as presented in the text are more like results from a note book and are difficult to digest. It should be more thoughtfully and more clearly written.

The second major point is that section 3.5 comes too early in the paper. The main focus of the paper is the relationship of the EUC to the Peruvian upwelling. This needs to be discussed very thoroughly before going on to the impacts on the fishery. Though the fishery is important as far as the ecosystem is concerned the science has to be dictated by the physics first, before jumping ahead to the biology of the system. The biology is a consequence of the physical oceanography as presented in this paper.

I would therefore suggest you consider a major revision of the structure of the paper. Eg Mean Annual cycle, Interannual variability, Discussion and Summary of your results regarding the EUC and upwelling. I am not sure the ecosystem discussion section 3.5 is necessary, but it could be added in the final conclusion on the impact of the upwelling on the fishery.

Minor points:

Page 1 Line 9 Should 8 deg S. ?

Page 2 Line 35 Replace "in" by "on"

Page 2 Line 56 remove first occurrence of "in"

Page 2-3 Line 53-60 Will need revising in light of my suggestions above.

Page 3 Line 80-81 The number of particles used to initialise the trajectories is remarkably variable. Need to explain carefully why this is the case and how this will affect your results.

P5 line 119 The clear seasonal variability is mentioned here and that is why I suggest you should discuss the mean annual cycle first before discussion of interannual variations. Eg Mean Annual Cycle only in sections 3.2, 3.3 and 3.4 As it stands the results in particular in 3.4 are presented in a confusing manner.

Page 8 Section 3.5 As stated above this is the application of the EUC to upwelling and should be placed after Section 3.6

Page 8 line 276-277 This is more than an inference. It is your main hypothesis and therefore should be stated clearly. Secondly it is stated the EUC is highly variable without stating the mean transport and its variability and interannual or seasonal?

---

## Author Response (AR1)

**Responses to Referees**

In the following responses to Referees 1-3, we reproduce each referee comments in black font, with our response in blue font and "changes made in the manuscript" in green font.
* * *
**Referee #1 : David Webb**

Overall this is a well presented and nicely written paper on the sources of upwelled water off Peru and Ecuador. It does this by following particles tracks from the upwelling regions and shows that much of the upwelled water comes from the Equatorial Undercurrent.
The authors show that the fluxes change from year to year and relate this one to the changing El Nino/La Nina was weak and needs to be improved.

Response: We thank the referee for some positive remarks.

Author's changes in manuscript: The major changes are twofold: extending particle trajectory calculations to monthly releases; diagnosing the upwelling associated with coastal winds (see below for details).

**Main points:**
Comment 1:
The paper only considers upwelling that occurs in the December of each year studied but do not give a reason for this. Figure 3 and Figure 4 indicate that December is a period of weak upwelling off Peru and Ecuador.  Was December chosen because it includes the height of the fishing season, is the time when large El Ninos have the most impact, or for some other reason?

Response: According to previous studies (Espinoza-Morriberon et al, 2017, and our results) releasing particles in the Peruvian upwelling system, the seasonal cycle of the vertical mass flux (referred to as the upwelling) peaks from August to October and weakest from December to February, in accordance with the wind stress and nitrate flux. We considered December by association with El Niño. On reflection, and in response also to Reviewer 2, we will extend our trajectory analysis to include particles released throughout the seasonal cycle.

Author's changes in manuscript: We now calculate and use trajectories based on monthly releases, for our analysis and discussions.

Comment 2:
The paper needs more details on the criteria used for the seeding. Was it for example one particle per cubic meter upwelled during each December?

Response: Thank you for your comment. On line 79, we specify 'Allocating particles in proportion to the upwelling rate', although we can provide more information. Specifically, in each grid cell, we allocate 1 particle per 10 m month$^{-1}$ of upwelling in excess of 50 m month$^{-1}$, per grid cell. For example, in the simple case that upwelling is 90 m month$^{-1}$, the excess upwelling rate is 40 m month$^{-1}$, so we allocate four particles to this grid cell, distributed spatially in a 2x2 array.

Author's changes in manuscript: We have added a more detailed explanation about the seeding off Peru in the methodology and caption of Figure 1, in which we illustrate the initial position of particles at 30 m and 100 m, for example years 1997 and 1998 (all months). See also lines 89-92.

Comment 3:
I find it difficult to reconcile figure 5 and 6, in which few particles travel 5 degrees eastward during December, with figure 10 b and d, which show 20% of particles travelling 70 degrees during the year, or figure 7 which implies that some particles come from beyond the dateline.

Response: Thank you for your comment. The EUC has been in-situ measured and modeled along the Pacific from 143E to 95W (Bryden, 1985; Johnson et al., 2002; Tsuchiya et al., 1989) and its termination off Peru and Ecuador to feed other major currents (Karnauskas et al., 2010; Lukas, 1986; Montes et al., 2010), since it was discovered in 1952 (Cromwell et al., 1954; Knauss, 1959). Year by year, strong changes in the EUC (velocity, core depth and transport, also including Kelvin waves) have been observed, however when El Niño happens, these changes are characterized by a significant eastward transport of water masses (warm pool). It is then possible for the EUC to extend all the way from the westernmost Pacific to the easternmost region, when its velocity exceeds a threshold (e.g., 1996-97 daily velocities in NOAA moored buoys, also shown in Figure 9 with transport values >+20 Sv). Under these circumstances, a considerable fraction of virtual particles is derived from the EUC, as is evident in this study.

Author's changes in manuscript: With monthly releases, we more completely track interannual changes in equatorial dynamics. To further explain how the particles back-track upwelled waters through the EUC at annual timescale, we cite additional references in the introduction and throughout the manuscript.

Comment 4:
Figure 7 needs more explanation, especially 7 a and b (where the caption should refer to the log scale and the base). I presume that "particle concentration" is not "the average of the particle concentrations at the start of each year" but is more "the average of the number of particles passing through each averaging box during the course of the year".

Response: Thank you for your comment. Your interpretation is correct, and we will provide more details. We can alternatively now refer to this diagnostic as 'fractional particle presence'.

Author's changes in manuscript: We have added more details about 'particle concentration' in the methodology (lines 98-102) and also in the caption of Figure 5.

I would also like to see a figure showing where particles started. Many obviously started in the regions where, on average, the age is greatest, but as the paper is really about where the upwelled water comes from, age by itself is not enough. Such figures might also help to clarify point 3 above.

Response: We provided this information already in Figure 5 and 6, where the initial positions are shown as black dots for each year. We will clarify the caption to emphasize this point.

Author's changes in manuscript: In Figure 1, we now show the initial location of particles for example experiments of 1997 and 1998.

Comment 5:
My main problems with this paper occur once it starts discussing year to year variations.
Around line 35 the paper discusses how the easterly trade winds generates a pressure head in the western Pacific and how below the surface the pressure gradient (high in the west, low in the east)

drives the undercurrent. Then around line 195 it discusses the flattening of the thermocline during the onset of an El Nino and that this "allowed more of these waters to progress all the way to the eastern boundary, where upwelling continued along the Peruvian coast. Conversely …"

To me this does not make sense because if the thermocline is flat, there is no east-west pressure gradient and no Undercurrent.

Response: Flattening of the thermocline during El Niño of 1997-98, due to the weakening (or reverse) of the trade winds in the western and central Pacific region (McPhaden, 1999) is typical of the ENSO variability seen in historical model simulations (Terada et al., 2020) and observations (NOAA buoy array data) (Kessler & Mcphaden, 1995). We do not claim that the thermocline is completely flattened. Rather, it flattens *to an extent*, shallowing in the western basin and deepening in the eastern region – warming up the cold tongue as observed in previous El Niños (Kessler & Mcphaden, 1995; McPhaden, 1999) for 1 to 3 months (observed in NOAA in situ data). Furthermore, we find that associated with flattening of the thermocline is eastward extension of the EUC, providing relatively more of the upwelling 'source waters' at the eastern boundary. We will clarify this point in the revised manuscript.

Author's changes in manuscript: In the introduction and also in our discussions, we now explain this relative flattening more carefully (lines 48-51, 221-222). We have added further references to previous studies that highlight this characteristic evolution of the equatorial thermocline and the EUC.

There are related problems with figure 10. Upwelling is largest in December 1997, at the height of one of the strongest El Ninos when the fraction of particles coming from the central pacific is also very large. The year 2000 has similar properties at the time of a weak La Nina, but almost nothing happened in 1998 and 1998 when there were strong La Nina. In 1992 there was also a reasonable El Nino, but with hardly any upwelling.

To me this means that simple arguments in the paper are not working – maybe scatter plots of upwelling volume against El Nino index or mean distance travelled in the Undercurrent against El Nino index would show something – but I suspect that more is needed.

Response: We will consider presenting the results obtained from releasing particles in other months throughout the years (see earlier response). Our findings will be accordingly updated.

Author's changes in manuscript: In Sect. 3.4 "EUC transport across the eastern Pacific and the variable EUC contribution to Peruvian upwelling", we clarify that these changes in transport relate to the *percentage* of particles found along the Pacific (lines after 203). We further account for upwelling that is more clearly attributed to coastal processes in the new sub-section 3.5 "The coastal wind-driven contribution to Peruvian upwelling". To clarify the variable partitioning of EUC and more locally sourced upwelling, we now include a schematic (Figure 9), to emphasize how upwelling may be associated with either process (see also lines after 253).

One of the problems appears to be the length of time integrated. Although the index indicated a large El Nino in December 1997, this was after a full year in which the El Nino was developing. Particles starting in the west or central Pacific early in the year would have travelled eastwards on a strong undercurrent. As the year developed the particles may have stayed ahead of the change in surface winds and so reached the western Pacific. If this is the case, then it was the strong undercurrent early in the year which carried the water eastwards to be upwelled, not the fact that El Nino index was large

in December. In addition – why, in the middle of a strong El Nino, when the trades had failed, why was there so much upwelling in the east?

Response: Upwelling associated with the EUC is larger (considering those particles sourced from the eastward-extended EUC), but not the absolute upwelling. So, the upwelling is weaker in 1997, just proportionally more from the EUC. As already mentioned, in revising the manuscript we will expand the Lagrangian analysis to back-track particles released in other months of the year, which should account for progressive changes in both the EUC and the coastal upwelling system during evolving ENSO events.

Author's changes in manuscript: Back-tracking particles throughout each year, we can now account for evolving ENSO events, and the manuscript has been revised accordingly. As previously mentioned, we now include a schematic representation (Figure 9) to graphically explain eastward extension of the EUC and associated upwelling close to the eastern boundary (see also sub-section 3.4).

There is also a problem with the year 2000 when there was a weak La Nina. Then both the undercurrent and the winds would have been strong – so by the normal theory all the undercurrent water would be expected to be upwelled before arriving off South America. So why was upwelling so strong this year and why was so much of it from the undercurrent?

Response: We will complement this point with our year-round particle releases. Although, we think that in the 'absence' of supply from the EUC, that coastal upwelling is sourced more locally. Also, EUC intensification not only occurs during El Nino events, but also during other warmer events such as in 2000.

Author's changes in manuscript: With monthly particle releases, we still obtain high % contribution of the EUC to upwelling in 2000 (Fig. 7c), while the upwelling flux is slightly weaker than the 1998-2007 average (see Fig. 8). In Sect. 3.4 "EUC transport across the eastern Pacific and the variable EUC contribution to Peruvian upwelling", we note the persistence of positive EUC transport anomalies into 2000, long after the La Niña of 1998/99 (lines 241-244).

Also why also was upwelling less in other weak La Nina years and why, when there were strong La Nina of 1998-1999, was there little upwelling and so little water coming from the central pacific?

Response: As explained previously, EUC waters are typically upwelling well to the west of the coastal upwelling zone in La Niña years, when the more strongly tilted thermocline brings EUC waters into the surface Ekman layer by the longitude of Galapagos. During a strong La Nina 1998-1999, the EUC next to Galapagos is almost absent (or in the eastern region), so Peruvian upwelled waters sourced from the EUC are consequently less than during weak La Nina. Only local water masses are feeding the Peruvian upwelling region (study area). As also mentioned, we will revisit this finding with more complete Lagrangian analyses (year-round releases). We will further develop and provide a metric of the Peruvian upwelling, 5-daily for the ORCA12 hindcast (1988-2007), alongside diagnostics of predicted coastal upwelling, based on Ekman theory, to which some of the variability in upwelling may be attributed.

Author's changes in manuscript: In Figure 8, it is now evident that upwelling strengthened in absolute terms after 2000. We further explain in lines 263-267 why in La Niña events there is a lower percentage of particles feeding the Peruvian system from the EUC, as a consequence of more wind-driven coastal upwelling (see also Figure 9).

Comment 6:
Another area in which I am unclear is the relation of El Ninos to fisheries. As I understand it the fisheries in the region are successful because the upwelled water is full of nutrients. This normally implies that it comes from deep in the ocean and that it has not recently been within the surface photic zone where it would lose nutrients. So Undercurrent water is just perfect.

I also understand that the reason research on the El Nino started was that with the relaxation of the trades during an El Nino, the surface mixed/nutrient poor layer became
thicker and any upwelled water was nutrient poor.

So how does this match with so much December 1997 water coming from the undercurrent? At the height of a strong El Nino the water in the photic zone above 100 m should have all spent some time near the ocean surface.

Response: El Niño can have positive and negative effects in the fisheries regarding the species we are analyzing (Ramiro Castillo et al., 1997; Raul Castillo, 1996; Contreras Paya, 2017; Icochea et al., 1989; Tam et al., 2008; Taylor et al., 2008). In this case, we discuss the results obtained from particles travelling within the EUC reaching the Peruvian coast, regarding one of the most abundant demersal species in Peru, Peruvian Hake, that has been widely studied as a bioindicator for the EUC strengthening off the Peruvian coast (Icochea et al., 1989; Tam et al., 2008; Taylor et al., 2008). Also, we have mentioned other benthic and pelagic species that were affected by the presence of the EUC before/during El Niño (Martina, 2004). Of course, the EUC is not the only variable influence on fisheries off Peru. Further analysis of variable coastal upwelling (see earlier comment) will put the EUC influence in a wider context. (We will add these extra references in our work)

Previous studies have established that the EUC brings important nutrient-rich waters to the southeastern part of our study region (Qin et al., 2016; Slemons et al., 2009; Vichi et al., 2008). Espinoza-Morriberon et al. (2017) explore the relation between vertical fluxes (upwelling) and nitrate with the ROMS-PISCES coupled model, finding that nutrients decrease dramatically during extreme El Niños and other warm events. We will cite and discuss this corroborative evidence in the revised manuscript.

Author's changes in manuscript: We have added supportive bibliography for discussing these changes in the EUC and how this affects water masses upwelling off Peru.

**Conclusions**
Given these problems I do not really want to spend time on other details. I think a full solution needs a lot more work, a better knowledge of the winds causing upwelling, the effect of stratification on the amount of upwelling, the changing strength of the undercurrent and upwelling at different longitudes and different times of year.
I do not think that I can ask for this to be done before publication but I see two ways forward:
The first is to accept the difficulties. The paper successfully shows that, on average, a large fraction of the upwelled water can come from the Undercurrent as opposed to currents running along the coastline. The variations with time can also be presented but with the difficulties pointed out and possible explanations noted. Papers that highlight problems with current ideas often get many citations.
Another possibility is to concentrate on just the last three months or so of each year, during which the El Nino index will not have changed so much. You should already have the data and it may be simpler to relate the amount of upwelling and contribution from the undercurrent (near the Galapagos) to the El Nino index and the winds near the coast.

Regards,
David Webb.

Response: Thank you for your valuable comments, we will extend our trajectory analysis to include particles released throughout the seasonal cycle, and separately examine the variations in coastal upwelling (attributed to variations in equatorward winds).

Author's changes in manuscript: With monthly releases, we obtained a more robust dataset for explaining the evolving EUC contribution to Peruvian upwelling. We also analyzed wind-driven coastal upwelling, to better identify the relative importance of changing coastal winds (Sect. 3.5). With these more comprehensive results, we expanded the bibliography accordingly.
* * *
**Editor Comment**

Comment 1:
There is an important omission in this paper and that there is no reference to a recent paper in JGR Oceans doi:10.1029/2020JC016609 by Karnauskas et al. entitled "The Pacific Equatorial in Three Generations of Global Climate Models and Glider Observations".

This should be referenced in your paper in both the bibliography and the main text.

Response: Thank you for your comment. We will consider this reference in our work.

Author's changes in manuscript: Karnauskas et al. (2020) is now cited at line 28.
* * *
**Referee #2 : Anonymous**

This study makes an attempt to link the interannual variability of the Equatorial Undercurrent (EUC) to that of the Peruvian upwelling, by back-tracking particles released in the Peruvian upwelling region to the equatorial Pacific. Particles tracking is an effective technique in tracing the origins of water masses. The application of the technique in this study, however, is insufficient. The authors estimated the contribution of the EUC to the Peruvian upwelling from just one release per year without demonstrating that this one release is representative of the oceanic condition of the corresponding year. For a fuller exploration of the connection between the EUC and the Peruvian upwelling, particles need to be released throughout of a year so that stable statistics can be obtained.

The analysis and interpretation of the results are also insufficient. Results from particle tracking show that strongest influence of the EUC to the Peruvian upwelling is in 1997 (an El Nino year). The author's explanation is flawed (e.g. flattening of thermocline, lines 201-203), or has no physical basis (e.g. lines 215-216). I suggest that the authors consider more carefully the timing of various events – the release time of particles, the transit time for particles to reach the equator, and the structure of the EUC near the eastern boundary at the time of particles' arrival so that a clearer view of particles dispersal can be obtained.

Response: Thank you for these suggestions, which are largely similar to those of Referee #2. As stated in previous response, extension of our trajectory analysis to include particles released throughout the seasonal cycle should address one concern, while a more careful explanation of how relative flattening

of the EUC (during El Niño) can supply a greater fraction of upwelling water off Peru may convince the reviewer of this point.

**Some Specific comments**

Comment 1:
 1. Introduction
Line 48:  Are there available data for the total pelagic fish landings in 1997, 1998 and 1999?

Response: Yes, there is available information on landings since 1980 up to now from the Peruvian Sea Institute (IMARPE - http://www.imarpe.gob.pe/imarpe/index2.php?id_seccion=I013102000000000000000 ) , although, data from 1980 to 1999 are only available as reports or published papers such the one we cited in our work (Ñiquen and Buchon 2004).

From those sources, we established that more than 90% of the total pelagic landings in Peru is represented by the Peruvian Anchovy (also here in Bouchon Corrales M., 2018 doctoral thesis https://rua.ua.es/dspace/bitstream/10045/103709/1/tesis_marilu_bouchon_corrales.pdf
  And https://revistas.imarpe.gob.pe/index.php/boletin/article/view/171/161 - Spanish version only, although there are tables and figures easy to follow).

That is why we used anchovy as the main pelagic species when explaining El Nino-EUC effects off Peru. We will add this last reference to our work.
In case of 1999 landings: (http://biblioimarpe.imarpe.gob.pe/bitstream/123456789/1827/1/INF%20155.pdf Spanish version only)

Author's changes in manuscript:  We have expanded bibliography in the introduction and part of discussion, for these species.

Comment 2:
 2. Methodology
Line 85 : Peruvian upwelling happens year-round with variability (your figure 3 and 4), what makes December 31 a good release time for particles to sample interannual variability?

Response: As for Reviewer 1 -  According to previous studies (Espinoza-Morriberon et al, 2017, and our results) releasing particles in the Peruvian upwelling system, the seasonal cycle of the vertical mass flux (referred to as the upwelling) peaks from August to October and weakest from December to February, in accordance with the wind stress and nitrate flux.

We considered December by association with El Niño. On reflection, and in response also to Reviewer 2, we will extend our trajectory analysis to include particles released throughout the seasonal cycle.

Author's changes in manuscript: Now with monthly releases (see responses to Referee #1), we obtain more robust data for explaining EUC dynamics. Correspondingly, we have expanded citations in the introduction and throughout the manuscript.

Comment 3:
3. Results and discussion

Line 172-173 : How did the authors determine from Fig. 8 that there was a flattening of the thermocline?

Response: We do not determine flattening of the thermocline directly from Fig. 8, where we use density on the y axis, rather refer to what is known. We will clarify the text accordingly. We will also plot temperature along the Equator in depth-longitude space (an additional figure), to make clearer this seasonal flattening.

Author's changes in manuscript:  We have re-organized our manuscript for a better understanding of our results. In this new version of our paper, and on reflection, we are not using this figure here.

Line 181-183 : These lines state that the EUC disappeared from the central Pacific in December 1997 – January 1998, and that the EUC transport anomalies exceeded -20Sv for much of 1997.

Response: We will moderate this statement to explain that the EUC was substantially weakened during the El Niño of 1997, most notably around November of that year (Johnson et al. 2000), consistent with highly negative transport anomalies in the NEMO-ORCA12 hindcast.

Author's changes in manuscript: We have clarified this point (lines 219-227).

Lines 196-198: How do particles near 160W in the EUC arrive at the Peruvian upwelling region in 1997 in large numbers if the EUC transport is reduced or absent.

Response: As emphasized in the previous response, while the EUC weakens or is even briefly absent, it may extend further to the east during El Niño – consistent with flattening of the thermocline (so fewer particles upwell to the west). This would explain why more particles reach the eastern boundary via the EUC during El Niño, a point we will clarify in the revised manuscript. As for reviewer 1, we will provide further evidence for this explanation with analysis of year-round releases.

Author's changes in manuscript: Assisted with the schematic of Figure 9, we explain "In Fig. 9b, the EUC shoals more gradually to the east, reaching the coastal upwelling zone, where weakened winds otherwise limit local upwelling." (lines 290-293). This explanation is reinforced at lines 356-362.

Line 200-203: Do the authors suggest that the EUC can persist when the thermocline (pycnocline) is flat?

Response: No, but we do not suggest that the thermocline is literally flat, rather flattened – see previous responses for elaboration on this issue.

Author's changes in manuscript: See previous response.

Line 203-218: Perhaps this is an issue with timing instead of specific types of El Nino or La Nina. For example, the transport at 160W in late 1998 is unlikely making an impact on particles releases in the Peruvian upwelling region on 31 December 1998. The coastal flow and the EUC in the vicinity of the eastern boundary are much more relevant for the initial dispersal of particles.

Response: The negative transport anomaly actually persists for the second half of 1998 (Fig. 9b); given typical EUC velocity of 50 cm s$^{-1}$, and the range of longitude (spanning 80°, or 8800 km), transit times range up to 204 days; on this basis, we will refer instead to EUC transport anomalies across the eastern equatorial Pacific during the second half of 1998.

Author's changes in manuscript: As we now analyse particles released throughout the year, and separately analyse the wind-driven part of Peruvian upwelling, the 'timing issue' is no longer an impediment to our interpretation.

Line 229: Was there a la Nina event in November 1993-March 1994? April-August 1998 was a transition period from El Nino to La Nina.

Response: Along the Peruvian Coast generally, after a strong EL Niño, occurs La Niña such as was observed after El Niño 1991 to summer 1993, and El Niño 1997 to June or July 1998.

Author's changes in manuscript: We now more clearly explain these events (In sub-section 3.4, also see lines 264-2667and 303-306).

Comment 4:
 4. Conclusions
First sentence: The investigation is not systematic because only water that upwells in the Peruvian region in December each year is tracked. No evidence is provided that December is representative of the whole calendar year.

Response: We considered December by association with El Niño. On reflection, and in response also to Reviewer 1, we will extend our trajectory analysis to include particles released throughout the seasonal cycle.

Author's changes in manuscript: All Lagrangian analyses are now based on monthly particle releases.

Lines 285-286: The EUC is driven by zonal pressure gradient. When the thermocline (pycnocline to be precise) is flattened at a certain longitude, does the EUC not weaken or disappear at that location.

Response: As for Reviewer 1 - Flattening of the thermocline during El Niño of 1997-98, due to the weakening (or reverse) of the trade winds in the western and central Pacific region (McPhaden, 1999) is typical of the ENSO variability seen in historical model simulations (Terada et al., 2020) and observations (NOAA buoy array data) (Kessler & Mcphaden, 1995). We do not claim that the thermocline is completely flattened. Rather, it quasi-flattens, shallowing in the western basin and deepening in the eastern region – warming up the cold tongue as observed in previous El Niños (Kessler & Mcphaden, 1995; McPhaden, 1999) for 1 to 3 months (observed in NOAA in situ data). Furthermore, we find that associated with flattening of the thermocline is eastward extension of the EUC, providing relatively more of the upwelling 'source waters' at the eastern boundary. We will clarify this point in the revised manuscript.

Author's changes in manuscript: As previously explained, we now clarify that while the EUC may be rapidly weakened in the Pacific (due to a rapid eastward transport) in warm events, relatively more particles might reach the eastern boundary to upwell, and vice-versa (lines 236-252).
* * *
**Referee #3 : Anonymous**

Major points:

This is an interesting paper addressing the relationship between the equatorial undercurrent to the upwelling off the coast of Peru.

The objectives of the paper are reasonably clear, in particular, the Lagrangian particle tracking as applied to 20 years of regional data from a high resolution global model to investigate the above relationship.

The focus of the paper is on interannual variability. Though I do realize the interannual variability is large in this region, I did think a more logical approach would be to discuss the mean annual cycle (20 year average of each month) first to establish particle tracking and its interpretation. Though this may have been covered in previous published studies your results are based on a very high resolution model and this is new. Therefore, it would be better to introduce these results first before the interannual results. The proposed section would lead the reader to an understanding of the particle tracking method and its interpretation. The results as presented in the text are more like results from a notebook and are difficult to digest. It should be more thoughtfully and more clearly written.

Response: Thank you for your valuable comment. We will improve the results section in a more systematic manner.

Author's changes in manuscript: We have re-organized the manuscript, considering the annual mean description first, before explaining about interannual variations – see Sect. 3.3 "Mean pathways at annual timescales".

The second major point is that section 3.5 comes too early in the paper. The main focus of the paper is the relationship of the EUC to the Peruvian upwelling. This needs to be discussed very thoroughly before going on to the impacts on the fishery. Though the fishery is important as far as the ecosystem is concerned the science has to be dictated by the physics first, before jumping ahead to the biology of the system. The biology is a consequence of the physical oceanography as presented in this paper.

Response: Thank you for your valuable comment. We will re-organize these sections as suggested.

Author's changes in manuscript: In the re-organized manuscript we mainly focus in the EUC links to the upwelling region. In Sect. 4 "Summary and Discussion", we now cover impacts of the EUC variability on the ecosystem, as suggested.

I would therefore suggest you consider a major revision of the structure of the paper. Eg. Mean annual cycle, interannual variability, Discussion and Summary of your results regarding the EUC and upwelling. I am not sure the ecosystem discussion section 3.5 is necessary, but it could be added in the final conclusion on the impact of the upwelling on the fishery.

Response: Thank you for your valuable comment. We will re-structure the paper accordingly.

Author's changes in manuscript: In particular, we have re-structured the Results section as follows:
3.1 Evaluation of NEMO-ORCA12 hindcast in the equatorial Pacific

As mentioned, discussion of EUC impacts on fisheries has now been moved into Sect. 4.

**Minor points**

Page 1 Line 9 Should 8 deg S.?
Response: Corrected.
Author's changes in manuscript: We have corrected this in our manuscript.

Page 2 Line 35 Replace "in" by "on"
Response: Corrected.
Author's changes in manuscript: We have modified it in our manuscript.

Page 2 Line 56 remove first occurrence of "in"
Response: Corrected.
Author's changes in manuscript: We have corrected this in our manuscript.

Page 2-3 Line 53-60 Will need revising in light of my suggestions above.
Response: We will modify it accordingly.
Author's changes in manuscript: We have revised this accordingly.

Page 3 Line 80-81 The number of particles used to initialize the trajectories is remarkably variable. Need to explain carefully why this is the case and how this will affect your results.

Response: The number of particles is proportional to the strength of upwelling (see response to Comment 2 of Reviewer 1).

Author's changes in manuscript: Using monthly releases, we now have 12 times more data than before, increasing robustness of our analysis. Moreover, we now clearly explain that the number of particles is proportional to the strength of upwelling.

P5 Line 119 The clear seasonal variability is mentioned here and that is why I suggest you should discuss the mean annual cycle first before discussion of interannual variations. Eg. Mean Annual Cycle only in section 3.2, 3.3 and 3.4 As it stands the results in particular in 3.4 are presented in a confusing manner.

Response: Thank you for comment. We will enhance and re-organize the results accordingly.

Author's changes in manuscript: We now describe and explain the mean cycle before interannual variations.

Page 8 Section 3.5 As stated above this is the application of the EUC to upwelling and should be placed after Section 3.6

Response: Thank you for comment. We will re-organize these sections accordingly.

Author's changes in manuscript: We have re-organized the manuscript in a clearer way (see earlier responses).

Page 8 line 276-277 This is more than an inference. It is your main hypothesis and therefore should be stated clearly. Secondly it is stated the EUC is highly variable without stating the mean transport and its variability and interannual or seasonal?

Response: We will state our hypothesis more clearly in the conclusions. We presented the mean transport values, interannual variability and seasonal variability in Figure 1, Figure 9, Table 1 and Table A3. However, we will improve the clarity and presentation of these results.

Author's changes in manuscript: We now state that "We specifically address the role of the EUC in Peruvian upwelling, relative to local wind-driven coastal upwelling in the zone 5-10°S. We aim to quantify absolute and relative changes in the provenance of waters upwelling off northern Peru, to establish the extent of interannual variability and links to ENSO events." (lines 56-62). We now provide the monthly mean EUC transport for a range of longitudes, in Table 1.

---

## Author Response (AR2)

**Final Revisions**
* * *
**Referee #1 : David Webb**

No minor revisions
* * *
**Referee #2 : Anonymous**

**Minor revisions:**
Review of "Interannual variability in contributions of the equatorial undercurrent (EUC) to Peruvian upwelling source water" by Gandy Maria Rosales Quintana et al.

I'm glad that the authors have conducted additional particle tracking experiments to address the previous lack of temporal sampling. The results from the additional experiments, however, do not appear to be utilized much in analysis.

An example: In Figure 5b (left and right) where particle age is displayed, the vertical axis is in calendar months. This is fine if only the release on 31 December of each year is considered. Now there are monthly releases, there is no longer a direct correspondence between age and a calendar month, what does this figure represent?

Response: This legend labelling was an oversight; we record age (prior to the start of backtracking in the coastal upwelling zone), irrespective of which month (Jan-Dec) that we start back-tracking. We have re-labelled the legend in days (0-365) prior to upwelling. Thank you for pointing this out.

Also, on lines 173-174: the text says "After around 2 months, particles are back-tracked to the Galapagos Islands …", the age contour of 180 days is near the islands, this is much longer than 2 months.

Response:  We appreciate your scrutiny. We intended to refer to the time elapsed between average age near the Galapagos (180 days) and the approximate 120-day age at the edge of the upwelling zone. We have modified the text as, "Back-tracked particles are located near the Galapagos Islands around 2 months prior to arrival at the outer limit of coastal upwelling (noting mean age increases from 120 to 180 days) … (Line 176-177)"

And further, on line 180: 60-90 days (October-September)? The release on 31 December only?

Response: In these lines, we are referring to our 12-month (year-round) releases, and again reference to October-September is an oversight. We should more simply explain how, on a timescale of 60-90 days, back-tracked particles are traced to somewhat greater depth. We now clarify this text as follows: (Line 184)"Near the eastern boundary, particles upwelling across 100 m are traced to depths below 125 m, around 60-90 days prior to upwelling, but particles back-tracked into the EUC remain in the mean depth range 100-125 m from the Galapagos Islands to around 120°W."

Another problem with the analysis is grouping the results into calendar years. This does not represent well El Niño and La Niña events that usually cross calendar years. Figure 6 shows that there is considerable seasonality as well as interannual variability in the EUC transport, in addition to longitudinal variations. For example, particles released in 1997 (31 January to 31 December, 12 releases) are influenced by the current system during the time period of 31 January 1996 to 31 December 1997. Are these particles summed together to represent 1997 in Figure 7?

Response: Thank you for your comment. We intend here to compare experiments collectively, summing the data from monthly releases that span 2 years per experiment, to emphasize changes happening during warm and cold events that typically straddle calendar years – as pointed out by the reviewer. For instance, in the case of the experiment labelled '1997', particles sample currents over 1996 and 1997. We have accordingly modified the caption in Fig. 7, consistent with sampling flow across two years (per experiment). For further clarification, we have modified the text at line 94: "Note that for releases during a given year, particles sample currents over two years, across a calendar year boundary. For instance, if we back track particles throughout 1997, particles sample currents throughout 1996 and 1997. In analyzing Lagrangian data for releases through a given year (e.g., 1997), we aggregate the data across all 12 months of releases, and refer to the experiment accordingly (e.g., 1996/97)". In the caption of Fig. 7, we further clarify, "In (a) and (b), the number of particles per experiment are plotted at calendar year boundaries. In (c) and (d), particle percentages are labelled by the two calendar years across which currents are sampled."

Lastly, I would like to suggest a modification to Figure 9.

Fig. 9a: say "weak EUC in the eastern Pacific", for example, 1998/1999 La Niña.
Fig. 9b: say "strong EUC in the eastern Pacific", for example, 1997/1998 El Nño.

Note that the EUC in the central equatorial Pacific (e.g. 160°W) is strong in 1998/1999 and almost absent in 1997/1998 (Figure 6).

Response: Thank you for your suggestion. These clarification are now added.

---

## Author Response (AR3)

**Final Revision N.2**
* * *
**Remarks from the preceding review file validation:**

With the next revision, please remove the text part "Copyright statement. TEXT" from page 1.

**Response: We have removed that part in our final version.**

**Please consider these 2 updates:**

**1.  In Figure 9 as follow:**

[Figure]

*Figure 9a - before - modification*          *Figure 9a – after - modification*

**2.  In Acknowledgements:**

Line 392:   …"We also thank to **KAKENHI (20K20634, 19H01965) Japanese project** for financially supporting this publication"…